# Modulation of the Bile Acid Enterohepatic Cycle by Intestinal Microbiota Alleviates Alcohol Liver Disease

**DOI:** 10.3390/cells11060968

**Published:** 2022-03-11

**Authors:** Dragos Ciocan, Madeleine Spatz, Nicolas Trainel, Kévin Hardonnière, Séverine Domenichini, Françoise Mercier-Nomé, Aurore Desmons, Lydie Humbert, Sylvère Durand, Guido Kroemer, Antonin Lamazière, Cindy Hugot, Gabriel Perlemuter, Anne-Marie Cassard

**Affiliations:** 1Inserm U996, Inflammation, Microbiome and Immunosurveillance, Université Paris-Saclay, 92140 Clamart, France; madeleine.spatz@hotmail.fr (M.S.); nicolas.trainel@universite-paris-saclay.fr (N.T.); kevin.hardonniere@universite-paris-saclay.fr (K.H.); cindy.hugo60@gmail.com (C.H.); gabriel.perlemuter@aphp.fr (G.P.); 2AP-HP, Hepatogastroenterology and Nutrition, Hôpital Antoine-Béclère, 92140 Clamart, France; 3Paris Center for Microbiome Medicine (PaCeMM) FHU, 75012 Paris, France; aurore.desmons@aphp.fr (A.D.); antonin.lamaziere@sorbonne-universite.fr (A.L.); 4Inserm, CNRS, Institut Paris Saclay d’Innovation Thérapeutique, Université Paris-Saclay, 92296 Chatenay-Malabry, France; severine.domenichini@universite-paris-saclay.fr (S.D.); francoise.mercier-nome@universite-paris-saclay.fr (F.M.-N.); 5INSERM, Centre de Recherche Saint-Antoine, CRSA, AP-HP. SU, Hôpital Saint Antoine, Département de Métabobolomique Clinique, Sorbonne Université, 75012 Paris, France; lydie.humbert@upmc.fr; 6Metabolomics and Cell Biology Platforms, Gustave Roussy Cancer Campus, 94800 Villejuif, France; sylvere.durand@gustaveroussy.fr (S.D.); kroemer@orange.fr (G.K.); 7Centre de Recherche des Cordeliers, INSERM, U1138, Equipe 11 Labellisée par la Ligue Nationale Contre le Cancer, 75005 Paris, France; 8Université Paris Descartes/Paris V, Sorbonne Paris Cité, 75005 Paris, France; 9Université Pierre et Marie Curie, 75006 Paris, France; 10Pôle de Biologie, Hôpital Européen Georges Pompidou, AP-HP, 75015 Paris, France

**Keywords:** alcohol, bile acid, microbiota, bacteria, liver, pectin, fiber

## Abstract

Reshaping the intestinal microbiota by the ingestion of fiber, such as pectin, improves alcohol-induced liver lesions in mice by modulating bacterial metabolites, including indoles, as well as bile acids (BAs). In this context, we aimed to elucidate how oral supplementation of pectin affects BA metabolism in alcohol-challenged mice receiving feces from patients with alcoholic hepatitis. Pectin reduced alcohol liver disease. This beneficial effect correlated with lower BA levels in the plasma and liver but higher levels in the caecum, suggesting that pectin stimulated BA excretion. Pectin modified the overall BA composition, favoring an augmentation in the proportion of hydrophilic forms in the liver, plasma, and gut. This effect was linked to an imbalance between hydrophobic and hydrophilic (less toxic) BAs in the gut. Pectin induced the enrichment of intestinal bacteria harboring genes that encode BA-metabolizing enzymes. The modulation of BA content by pectin inhibited farnesoid X receptor signaling in the ileum and the subsequent upregulation of Cyp7a1 in the liver. Despite an increase in BA synthesis, pectin reduced BA serum levels by promoting their intestinal excretion. In conclusion, pectin alleviates alcohol liver disease by modifying the BA cycle through effects on the intestinal microbiota and enhanced BA excretion.

## 1. Introduction

Alcohol liver disease (ALD) is the first cause of chronic liver disorders in Western countries [1]. More than 2 million people had alcohol-associated cirrhosis in the U.S., and ALD is the first cause of liver-related mortality and of liver transplant indication [2]. ALD includes various histopathological characteristics, ranging from simple steatosis to steatohepatitis, fibrosis, and cirrhosis, but only 10 to 20% of patients evolve to advanced fibrosis and its complications [3]. In addition, certain ALD patients develop alcoholic hepatitis, with episodes of acute liver inflammation associated with high mortality and few therapeutic options [4]. Some factors, such as female sex and changes in the composition of the intestinal microbiota (IM) and microbial metabolism, contribute to susceptibility to ALD progression [4,5]. 

The IM contributes to host tissue homeostasis through the production of a large panel of metabolites, including bile acids (BAs), short-chain fatty acids (SCFAs), and amino-acid-derived metabolites [4]. BAs have been shown to be involved in the pathogenesis of several liver diseases, including ALD [4,6]. Primary BAs are synthesized and conjugated in hepatocytes before their release into the gut. As BAs possess intrinsic membranolytic properties and can cause senescence or cell death, they are toxic for both hepatocytes and bacteria. Conjugation with taurine or glycine renders them more hydrophilic and less toxic. Primary conjugated hydrophilic BAs are reabsorbed from the terminal ileum. However, bacterial bile-salt hydrolases (BSHs) deconjugate primary BAs, preventing their reabsorption. Many bacteria express BSHs, including Clostridium, Lactobacillus, Bifidobacterium, Eubacterium, Escherichia, and Bacteroides [7]. Consequently, unabsorbed BAs are transformed by bacteria into secondary BAs through hydroxylation, epimerization, esterification, and desulfation reactions [7]. BAs facilitate the absorption of dietary lipids and lipid-soluble nutrients [8] and act as signaling molecules by binding to host receptors, including the farnesoid-X receptor (FXR) and Takeda G-protein-coupled receptors (TGR5, also known as GPBAR1) [8]. BAs modulate lipid and energy metabolism through FXR activation in the liver, as well as their own biosynthesis, by a negative feedback loop following ileal FXR activation [8]. The role of FXR in several liver diseases has been studied in some detail [9,10]. FXR deficiency worsens liver injury in alcohol-fed mice [11]. Conversely, ileum FXR activation has a protective effect in ALD [12]. TGR5 is involved in a wide variety of functions, including metabolic expenditure, the inflammatory response, gut motility, and gallbladder homeostasis [13]. TGR5 deficiency worsens ALD, partially due to changes in the IM and BA pool [14]. As both FXR and TGR show distinct affinities for different BAs, FXR and TGR5 signaling are strongly influenced by changes in the IM and BA pool. After chronic ethanol administration, total plasma and hepatic bile-acid concentrations increase and likely synergize with ethanol to cause hepatocyte death. Patients who develop life-threatening forms of ALD, such as severe alcoholic hepatitis (sAH), manifest specific changes in their BA pool with a shift towards more hydrophobic and toxic species that may be responsible for specific changes in IM composition. Indeed, in these patients there is an increase in the plasma levels of total bile acids, plasma-conjugated cholic and chenodeoxycholic acid, ursodeoxycholic and lithocholic acid, whereas deoxycholic acid decreased [6,15]. In the feces ALD is associated with a decrease in total BAs and the secondary deoxycholic acid and lithocholic acid [6]. Moreover, it has been suggested that some BAs (taurochenodeoxycholic acid and tauroursodeoxycholic acid) can be related to disease severity [6,15].

Conversely, as previously discussed, the IM may also alter the BA pool by converting primary BAs to secondary BAs, resulting in a vicious cycle [15,16]. Moreover, intestinal concentrations of BAs can discriminate mice receiving feces from patients with severe ALD from animals receiving feces from individuals without ALD [17]. ALD is associated with alterations in the composition and function of the IM in animal models and humans [4,18]. Indeed, the fecal abundance of multiple bacteria expressing 3α, 3β, 7α, and 7β epimerases, enzymes involved in BA metabolism, are decreased in AH [6]. Interventions on the IM by fecal microbial transfer and prebiotic or probiotic treatments alleviate ALD in animal models. Microbiota-editing strategies could therefore constitute a promising therapeutic approach for the treatment of ALD [18]. Dietary soluble fiber is a means to increase BA clearance through the gut [19]. We have previously shown that oral supplementation with pectin, which can be considered a prebiotic, alleviates ALD by increasing bacterial production of indole derivatives, leading to AHR activation [17,20]. However, only a part of the beneficial effects of pectin were mediated by aryl hydrocarbon receptors. Therefore, other molecular mechanisms are likely involved in its beneficial effects.

Due to its physicochemical properties, fiber, such as pectin, can form a viscous matrix with cholesterol and BAs in the intestinal lumen, resulting in reduced systemic levels of cholesterol and BAs [21,22] and changes in glucose metabolism [23]. We, therefore, studied the effect of pectin on the enterohepatic BA cycle in a humanized animal model of ALD alcohol-fed mice treated with pectin. 

## 2. Materials and Methods

Mice. We used seven-week-old female C57BL/6J mice (Janvier laboratory, Le Genest, France). As female sex is an independent risk factor for ALD in both mice and humans, we chose to perform our experiment in female mice [3]. The research was conducted according to a protocol reviewed and approved by the French Ministry, 2015052715405651v2 (APAFIS#729) and 2017042314557080v1 (APAFIS#4788). 

Chronic exposure to alcohol. To induce ALD, we used the Lieber DeCarli (LDC) diet for 21 days, as previously described [20,24] (Ssniff, Spezialdiäten GmbH, Soest, Germany). Ethanol accounted for 28% of the total caloric intake. The control diet was an isocaloric amount of maltodextrin (Maldex 170, Safe, France). Pectin was given at the beginning of alcohol intake using an alternative LDC diet containing 6.5% pectin from apples (Sigma-Aldrich, Saint Quentin Fallavier, France). During administration of the LDC diet, mice did not have access to drinking water. Consumption was recorded and was similar between the alcohol-fed groups (data not shown).

Fecal microbiota transfer. Mice received feces from an alcoholic patient with biopsy proven severe alcoholic hepatitis (Human FMT), as previously described [17,25]. Before the FMT, mice were fasted for 1 h and given oral-gastric gavage with PEG (polyethylene glycol, Macrogol 4000, Fortrans, Ipsen Pharma, Boulogne-Billancourt, France). After four hours, human feces were administered by oral gastric gavage, as previously published [17,25].

Tissues and samples. Blood was collected for biochemical analysis (alanine aminotransferase, ALT, and BAs). Liver, proximal ileum, colon and brown adipose tissue were harvested for histology and RNA extraction, as previously described [14,20]. RNA quantification was performed using the same methodology as previously described by our group [14,20]. The oligonucleotides used in this study are available in the Appendix A. The caecal content was collected and frozen for caecal BA measurement and untargeted metabolomics. Fecal samples were collected from mice two days before euthanasia for gut microbiota analysis. In addition, liver samples were used for bile-acid and triglyceride measurements. 

Histology. Oil-Red-O staining was performed on 7-μm thick frozen liver sections. Immunohistochemistry staining was performed on 3 μm BAT paraffin sections using a mAb against UCP1 (Bio-Rad, Marnes-la-Coquette, France). Slides were scanned (NanoZoomer 2.0-RS digital slide scanner, Hamamatsu, Japan) and images were digitally captured (NDP.view2 software, Hamamatsu, Japan).

Bile acid quantification. BAs were measured in the plasma, liver, and caecum by high-performance liquid chromatography-tandem mass spectrometry, as previously described [26].

Analysis of fecal metabolites. Untargeted fecal metabolites were measured using gas chromatography coupled to a triple quadrupole mass spectrometer, as previously described [16].

Microbiota. Bacterial DNA was extracted from feces using a Qiagen QIAamp DNA Stool Mini Kit (Courtaboeuf, France) and the fecal microbiota was analyzed using Illumina MiSeq technology targeting the 16S ribosomal DNA V3-V4 region in paired-end mode (2 × 300 base pair) (GenoToul, Toulouse, France), as previously described [27]. The mean number of quality-controlled reads was 18,302 ± 7625 (mean ± SD) per mouse. Microbiome analyses was performed with QIIME and the functional composition of the intestinal metagenome was predicted using phylogenetic investigation of communities by reconstruction of unobserved states (PICRUSt) [28] and was accessed online (http://huttenhower.sph. harvard.edu/galaxy/, accessed on 26 November 2021).

Statistical analysis. Results are shown as the mean ± SEM. Comparisons were performed using ANOVA or the nonparametric Kruskall–Wallis test with Tukey’s or Dunn’s multiple comparison post-hoc test, as appropriate (Graphpad Prism 9.3, Graphpad Software Inc, La Jolla, CA, USA); *p* < 0.05 was considered to be statistically significant.

## 3. Results

### 3.1. Pectin Alters the Enterohepatic Cycle of Bile Acids in Alcohol-Fed Mice 

Fecal microbial transfer (FMT) from patients with severe alcoholic hepatitis (sAH) to mice worsens ALD in the Lieber DeCarli (LDC) model of ALD [17,20]. This was confirmed by higher ALT levels (Figure 1A), steatosis (Figure 1B, C), and hepatic mRNA levels of pro-inflammatory genes (Figure 1D) in mice that first underwent FMT from an sAH patient. Alcohol also induced more severe liver lesions in the same mice. Moreover, reshaping the IM by the ingestion of pectin, a soluble fiber, reduced ALD (Figure 1A–D), as we previously reported [17,20].

Mass spectrometric metabolomics showed changes in the overall composition of the BA pool in the plasma (Figure 2A), liver (Figure 2B), and caecum (Figure 2C) following pectin treatment of mice relative to that of the control and alcohol-fed conventional and FMT mice. Indeed, the alcohol diet resulted in higher total BA (TBA) levels in the plasma of alcohol-treated conventional mice (Figure 2A) and the livers of alcohol-fed FMT mice than control mice (Figure 2B) but lower levels in the caecum of both conventional and FMT alcohol-fed mice (Figure 2C). Conversely, pectin treatment resulted in lower plasma TBA levels than in Alc mice that did not receive pectin (Figure 2A) and showed a tendency towards lower plasma (Figure 2B) and liver TBA (Figure 2C) and higher caecal TBA (Figure 2C) levels as compared to alcohol-fed FMT mice, without reaching statistical significance. The proportion of conjugated and unconjugated BAs was also altered. Indeed, the alcohol diet resulted in a higher level of primary conjugated BAs, mainly TCA, in the plasma (Figure 2D and Figure 3A) and livers (Figure 2E and Figure 3B) of both conventional and FMT-treated Alc mice than in the controls (Figure 2D,E). 

Conversely, pectin treatment resulted in lower primary conjugated BA levels in the plasma (TCA and TMCA) than in conventional and FMT-treated Alc mice, whereas they remained as high in the controls (Figure 2D and Figure 3A). In the caecum, pectin treatment resulted in higher primary unconjugated BA levels (CA and CDCA) than in conventional and FMT-treated Alc mice, and there was a further decrease in secondary unconjugated BA levels (DCA, LCA, UDCA, wMCA) (Figure 2F and Figure 3C). Pectin treatment also resulted in higher primary conjugated BA (TCA, TMCA) and TUDCA levels in the caecum than in all other groups, notably FMT-treated Alc mice (Figure 2F and Figure 3C). Overall, pectin decreased plasma and liver BA levels, increased caecal BA levels, mainly those of primary BAs, CA, and MCA, and changed the overall composition of the BA pool towards more hydrophilic forms (Figure 3D,E). 

### 3.2. Bile Acid-Metabolizing Bacteria Are Enriched in Pectin-Treated Mice

Pectin can modify both the composition of the IM and induce an increase in BA excretion in the cecum, which affects the bacterial composition of the IM. Thus, we next studied possible changes in the abundance of BA-metabolizing bacteria, focusing on BSH, the enzyme responsible for the deconjugation (deamidation) of conjugated (amidated) BAs and those that catalyze the 7a/b-dehydroxylation required for UDCA production [29]. Among bacteria with BSH activity, pectin treatment resulted in a higher abundance of Bacteroides and Enterobacteriacae (Figure 4A), which are among the dominant members of the IM, than in untreated mice, whereas it resulted in a lower abundance of the number of subdominant bacteria with BSH activity, such as Lactobacillus and Enterococcus (Figure 4B). Metagenomic prediction algorithms indicated an overall increase in the abundance of BSH genes (Figure 4C). Although, there were no statistically significant differences in the abundance of bacteria with 7-HSDH activity in the guts of pectin-treated mice relative to that in the guts of alcohol-fed mice (Figure 4D), the overall gene prediction suggested that pectin raised the abundance of these genes (Figure 4E). This is consistent with the greater UDCA content observed in the caecum (Figure 3C).

### 3.3. Pectin Modifies Bile-Acid Signaling in the Gut, Liver, and Brown Adipose Tissue 

BA levels depend not only on their excretion by the liver but also on intestinal reabsorption, which is achieved both by passive diffusion and active transport, mainly in the ileum, before returning to the liver via the portal vein. The alcohol diet resulted in higher mRNA levels of multidrug resistance-related protein 3 (MRP3), which transports BAs into circulation, in hepatocytes (Figure 5A). This is consistent with the lower TBA excretion in the caecum and higher TBA levels observed in the plasma (Figure 2A,C). In the ileum, pectin treatment resulted in lower mRNA levels of MRP2, which excretes BAs, than those found in all the other groups, as well as those of IBAP, which transports BAs through enterocytes. Moreover, pectin treatment resulted in higher MRP3 mRNA levels in the ileum than in conventional and FMT-treated Alc mice (Figure 5B). In the colon, pectin treatment also resulted in higher mRNA levels of ASBT, which takes part in the reabsorption of BAs, than in conventional and FMT-treated Alc mice, as well as those of OST, which transfers them into circulation, which were higher than in conventional Alc and control mice (Figure 5C). 

Pectin also modified carbohydrate and lipid absorption. In the ileum, pectin treatment resulted in lower mRNA levels of the sodium/glucose cotransporter 1 (SGLT1), the transporter that contributes to glucose reabsorption, than in all other groups, although the glucose transporter 5 (Glut5) mRNA levels remained unchanged (Figure 6A). The mRNA levels of Glut2, which transfers glucose to the blood, were lower in pectin-treated mice than FMT alcohol-fed mice (Figure 6A). Pectin and microbiota-related changes led to modifications in the composition of glucose-related metabolites (Figure 6B), with lower levels of several small carbohydrates in the feces (Figure 6C). Moreover, pectin treatment also resulted in lower mRNA levels of ileum cluster of differentiation 36 (CD36) and the fatty acid binding protein (Fabp1), which are both involved in the intestinal uptake of fatty acids, than in all other groups (Figure 6D) and changes in the composition of lipid metabolites in the feces (Figure 6E,F). 

We next examined the FXR pathway in the gut. In mice treated with pectin, we observed less ileal expression of FGF15, a gene induced by FXR activation, than in all other groups (Figure 7B). Cyp7A1 and Cyp27A1 mRNA levels were higher than in all other groups and may reflect abrogation of the negative feed-back for both the classical and alternative liver pathways involved in BA synthesis due to decreased FGF expression (Figure 7A). This is consistent with the higher level of BAs observed in the caecum (Figure 2C). Pectin treatment also resulted in higher mRNA levels of FXR in the liver and colon than in alcohol-fed conventional mice, whereas there was no impact on FXR mRNA levels in the ileum (Figure 7B,C). TGR5 mRNA levels were also higher in the colon but not the ileum of pectin-treated mice than in all other groups (Figure 7B,C). 

TGR5 is expressed in brown adipose tissue (BAT) and mediates its activation [30]. Alcohol-fed mice are lean and have little white adipose tissue (WAT) but conserve BAT, which is involved in thermogenesis. Moreover, BAT activity can modulate ALD [31,32]. Indeed, BAT appeared to be activated, as suggested by the relative absence of surrounding WAT in alcohol-fed mice compared to the control and pectin-treated mice (data not shown). Histological analyses showed fewer lipid droplets and immunohistochemistry markedly higher levels of UCP1 in alcohol-fed mice than the control or pectin-treated mice (Figure 8A,B). BAT activation was previously shown to be associated with altered TGR5 receptor expression [30]. However, we did not find any effect of pectin on TGR5 mRNA levels in the BAT of alcohol-fed mice (Figure 8C).

## 4. Discussion

In the present study, we analyzed how pectin, a fiber that modifies intestinal microbiota and alleviates ALD, modifies the enterohepatic cycle of BA, in alcohol-challenged mice receiving feces from patients with alcoholic hepatitis. We show that pectin enhances BA excretion and changes the overall BA composition, favoring the increase of hydrophilic BAs (less toxic) in the liver, plasma, and gut. This effect was due to an enrichment of intestinal bacteria harboring genes that encode BA-metabolizing enzymes. 

Total plasma and liver BA levels were elevated after chronic ethanol feeding, a phenomenon that likely synergizes with ethanol to cause hepatocyte death. Nevertheless, previous studies in animal models of ALD have shown that FXR activation by FXR agonists, including obeticholic acid and fexaramine, or an FGF15 orthologue, decreased ethanol-induced liver injury, steatosis, and inflammation by decreasing hepatic expression of Cyp7a1 and modulating lipid metabolism without any significant changes in the BA pool [12,33]. Here, we observed that, instead of directly modulating the FXR/FGF-15/19 pathway, pectin treatment rather acts as an intestinal BA sequestrant. Indeed, BA sequestrants, such as cholestyramine, colestipol, and colesevelam, are already used in clinical practice as medications to reduce LDL cholesterol in combination with dietary modifications or statins or to increase glycemic control in association with antidiabetic treatments [34,35,36]. Recently, the use of colesevelam, a BA sequestrant, was reported to improve alcohol-and nonalcoholic-induced liver steatosis and to reduce hepatic expression of the Cyp7a1 protein, although without any tangible effect on liver inflammation [37,38]. Moreover, colesevelam was found to augment serum ALT levels in ethanol-fed mice, probably because this resin neutralizes other beneficial molecules, including free amino acids [39]. Beyond its function as a BA sequestrant, pectin modulates the fecal metabolome, thus increasing the generation of beneficial bacterial metabolites, such as tryptophan derivatives, that alleviate ALD [20]. Colesevelam inhibits hepatic Cyp7a1, whereas pectin appears to favor BA synthesis through the downregulation of the FXR-FGF15 axis. However, this increase is also accompanied by an increase in BA excretion into the feces, with a shift towards a more hydrophilic BA pool. This is consistent with the results of several clinical studies on various medical conditions (non-alcoholic fatty liver disease, cholestatic pruritus, Crohn’s disease, hypercholesteremia, and type-2 diabetes mellitus) showing that colesevelam increases BA synthesis [40,41,42,43,44].

Alcohol intake induces changes in the expression of BA transporters, including the upregulation of hepatic BSEP and MRP2 and both hepatic and ileal ASBT [45]. In addition, here, we report an increase in the level of MRP3, which is responsible for the alternative basolateral efflux system in hepatocytes. This could explain the increase in TBA levels in the plasma and the depletion of caecal BAs observed in alcohol-treated mice. MRP3 expression plays an essential role in protective and adaptive responses in the presence of BA overload, such as cholestasis, which is generally observed in ALD [15]. Although pectin had little effect on the expression of BA transporters in the liver, it modified the expression of BA transporters in the ileum and colon. Pectin notably induced decreased ileal expression of IBAP and MRP2. This may be due to disrupted FXR signaling, as FXR is known to induce I-BABP and MRP2 expression [46]. BAs are actively taken up in the terminal ileum via the ASBT transporter [47]. Pectin had no effect on ileal ASBT expression but increased its colonic expression. Recently, it was shown that blocking intestinal BA reabsorption using a gut-restricted ASBT inhibitor attenuated hepatic steatosis and liver injury in a chronic-plus-binge ALD mouse model. This effect was associated with increased hepatic Cyp7a1 expression and decreased ileal FXR activity [48]. 

Pectin had little effect on BA reabsorption due to its sequestrant properties. Moreover, as for the ASBT inhibitor, it decreased ileal FXR activation and increased hepatic Cyp7a1 expression. Of note, pectin was also shown to suppress intestinal BA absorption by increasing fecal BA elimination by up to 110% in high cholesterol-fed mice, thus decreasing the pool of ileal BA by 47% and that of total BA by 36% and downregulating the FXR-FGF15 axis [49].

BAs are also important regulators of the composition of the gut microbiome. In animal models of ALD, alcohol-associated changes in the IM were associated with increased levels of bacterial BSH, which deconjugates BAs [12]. Moreover, changing the composition of the IM using antibiotics was shown to attenuate hepatic Cyp7a1 expression and reduce ALD in mice, suggesting that increased BA synthesis depends on gut bacteria [12]. 

Patients with alcoholic hepatitis also show a decrease in fecal BAs accompanied by a decrease in unconjugated and secondary BA levels [15], whereas alcoholic patients with cirrhosis have been reported to show a higher fecal concentration of hepatotoxic DCA and higher secondary to primary BA ratios than healthy controls [50]. In our model, pectin led to an increase in conjugated BA levels in the gut and, notably, a decrease in the level of hydrophobic (highly cytotoxic) species, such as DCA, and an increase in those of hydrophilic (less toxic) BAs, such as UDCA. Pectin induced an increase in BSH and HSDH activity in the IM. However, it has been suggested that BSH genes are involved in the development of alcohol liver disease [12]. Indeed, germ-free mice transplanted with stool from alcoholic patients with cirrhosis showed a higher ability to deconjugate BAs and produced a higher level of fecal secondary BAs than germ-free mice receiving stool from healthy subjects. Nevertheless, the secondary BA, UDCA, produced by intestinal bacteria through a series of deconjugation and epimerization reactions of CDCA, alleviates ethanol-induced hepatocyte injury in vitro [51]. Therefore, although pectin treatment led to an increase in BSH levels, the IM shifted towards bacteria that produce hydrophilic secondary BAs, such as UDCA.

Chronic alcohol consumption is associated with decreased BAT content in mice, as well as impaired thermoregulation [31]. Moreover, physiological levels of circulating BAs can promote heat production in the BAT by increasing energy expenditure through the activation of uncoupling protein 1 (UCP1), which can be regulated by many factors, including BAs, via the TGR5-signaling pathway [30,52]. However, we observed a pectin-induced decrease in UCP1 expression in BAT. This could be related to a reduction in TGR5 agonists (mainly LCA and DCA) in the pectin group, although there was no difference in the expression of TGR5. If the depletion of UCP1 is associated with liver injury in ALD mice [32], our data would suggest that enhanced BAT function cannot explain the hepatoprotective effects of pectin. 

## 5. Conclusions

In conclusion, pectin alleviates ALD by modifying the IM and its metabolic capacity, including its ability to transform BAs, thereby altering the BA enterohepatic cycle (Figure 9). Pectin alters the composition and function of the IM, resulting in beneficial effects on BA metabolism. Although targeting specific pathways related to IM alterations and microbial-derived metabolites in ALD has shown promising results, the broader spectrum effects of pectin could be of great interest. As alcoholic patients typically consume little fiber, including pectin, the dietary supplementation of fiber as a prebiotic could offer a promising option for ALD management. Nonetheless, clinical trials are required to confirm its beneficial effects on ALD in humans.

## Figures and Tables

**Figure 1 cells-11-00968-f001:**
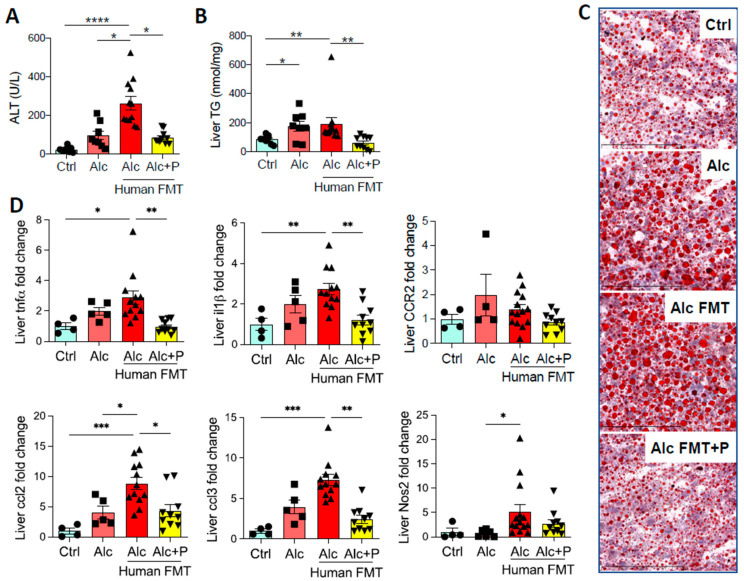
Liver injury in control and alcohol-fed mice and pectin-treated alcohol-fed mice. Control (Ctrl) and Alcohol-fed (Alc) mice received a Lieber DeCarli diet with isocaloric maltodextrin or alcohol, respectively. Pectin-treated mice (Alc + P) received 6.5% pectin in addition to 5% ethanol. Human fecal microbiota transfer (FMT) groups received microbiota from a patient with alcoholic hepatitis. (**A**) Plasma alanine transaminase (ALT) levels. (**B**) Liver triglyceride (TG) levels. (**C**) Representative histological images of Oil-Red-O staining of the liver (scale bar: Oil-Red-O = 100 µm). (**D**) Liver mRNA levels of pro-inflammatory cytokines and chemokines (tumor necrosis factor α, tnfα; interleukin 1 β, il1β; C-C chemokine receptor type 2, CCR2; chemokine (C-C motif) ligand 2, ccl2; chemokine (C-C motif) ligand 3, ccl3; Nitric oxide synthase 2, Nos2). * *p* < 0.05, ** *p* < 0.01, *** *p* < 0.001, **** *p* < 0.0001.

**Figure 2 cells-11-00968-f002:**
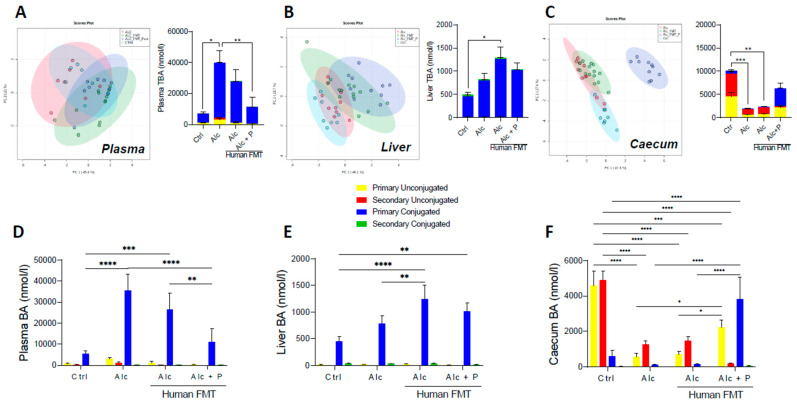
Pectin alters the bile-acid (BA) pool in different compartments. Control (Ctrl) and alcohol-fed (Alc) mice received a Lieber DeCarli diet with isocaloric maltodextrin or alcohol, respectively. Pectin-treated mice (Alc + *p*) received 6.5% pectin in addition to 5% ethanol. Human fecal microbiota transfer (FMT) groups received microbiota from a patient with alcoholic hepatitis. PCA ordination plot on all BA and total BA (TBA) data in the plasma (**A**), the first component explained 45.6% of variance and the second one 21%, liver (**B**), the first component explained 40.1% of variance and the second one 23.7%, and caecum (**C**), the first component explained 47.3% of variance and the second one 27.4%. Light blue-Ctrl, red-Alc, green-Alc FMT, dark blue-Alc + *p*. Quantification of primary and secondary conjugated and unconjugated BAs in the plasma (**D**), liver (**E**), and caecum (**F**). * *p* < 0.05, ** *p* < 0.01, *** *p* < 0.001, **** *p* < 0.0001.

**Figure 3 cells-11-00968-f003:**
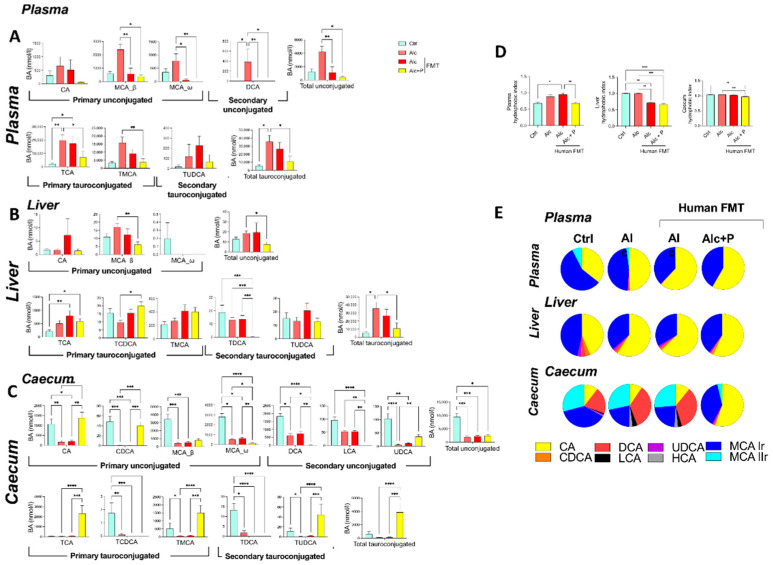
Pectin alters the bile-acid (BA) composition in alcohol-fed mice. Control (Ctrl) and alcohol-fed (Alc) mice received a Lieber DeCarli diet with isocaloric maltodextrin or alcohol, respectively. Pectin-treated mice (Alc + *p*) received 6.5% pectin in addition to 5% ethanol. Human fecal microbiota transfer (FMT) groups received fecal microbial transplantations from a patient with alcoholic hepatitis. BA composition in the plasma (**A**), liver (**B**), and caecum (**C**), hydrophobic index (**D**), and the relative composition of the bile pool (**E**). * *p* < 0.05, ** *p* < 0.01, *** *p* < 0.001, **** *p* < 0.0001.

**Figure 4 cells-11-00968-f004:**
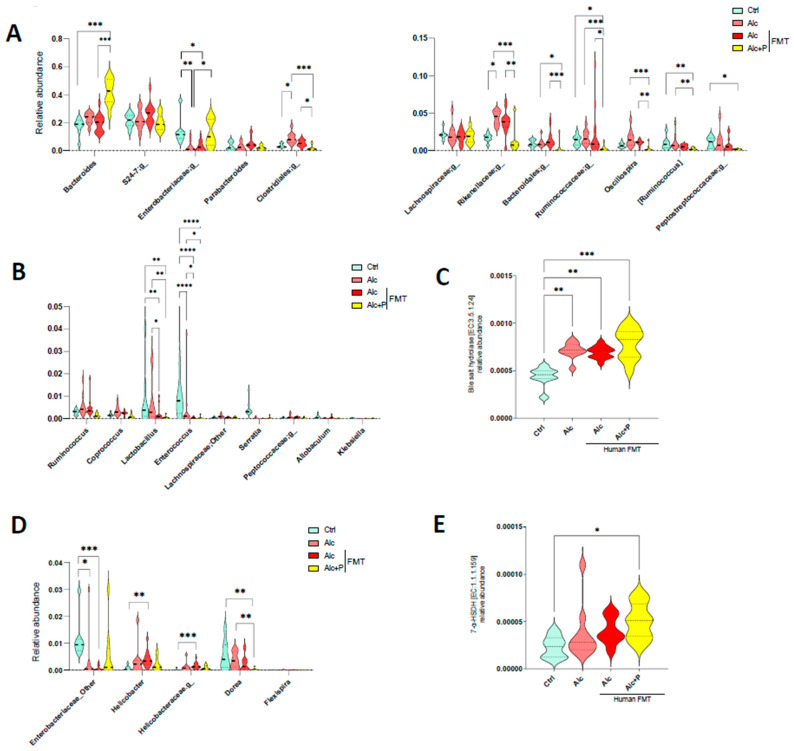
Alterations in the composition of bile acid-metabolizing bacteria in alcohol fed-mice. Control (Ctrl) and alcohol-fed (Alc) mice received a Lieber DeCarli diet with isocaloric maltodextrin or alcohol, respectively. Pectin-treated mice (Alc + *p*) received 6.5% pectin in addition to 5% ethanol. Human fecal microbiota transfer (FMT) groups received fecal microbial transplantations from a patient with alcoholic hepatitis. (**A**,**B**,**D**) Violin plots showing the relative abundance of bacteria with bile salt hydrolase (**A**,**B**) and 7 alpha-hydroxysteroid dehydrogenase (7-a-HSDH) activity (**D**) in all groups of mice (FDR *p* < 0.05). Predictive expression of bile salt hydrolase (**C**) and 7-a-HSDH (**E**). * *p* < 0.05, ** *p* <0.01, *** *p* < 0.001, **** *p* < 0.0001.

**Figure 5 cells-11-00968-f005:**
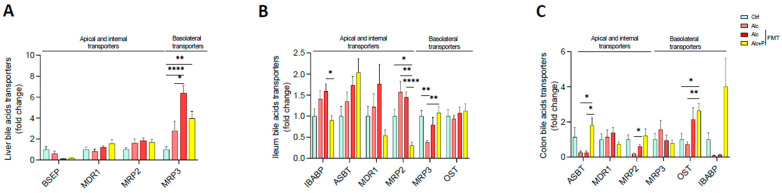
Alterations in bile-acid transport induced by pectin. Control (Ctrl) and alcohol-fed (Alc) mice received a Lieber DeCarli diet with isocaloric maltodextrin or alcohol, respectively. Pectin-treated mice (Alc + *p*) received 6.5% pectin in addition to 5% ethanol. Human fecal microbiota transfer (FMT) groups received the microbiota of a patient with alcoholic hepatitis. Quantification of mRNA levels of BA transporters (bile salt export pump, BSEP; multidrug resistance protein 1, MDR1; multidrug resistance-associated protein 2, MRP2 and 3, MRP3; ileal bile acid binding protein, IBABP; apical sodium-dependent bile salt transporter, ASBT; organic solute and steroid transporter, OST) by qPCR in the liver (**A**), ileum (**B**), and colon (**C**). * *p* < 0.05, ** *p* < 0.01, **** *p* < 0.0001.

**Figure 6 cells-11-00968-f006:**
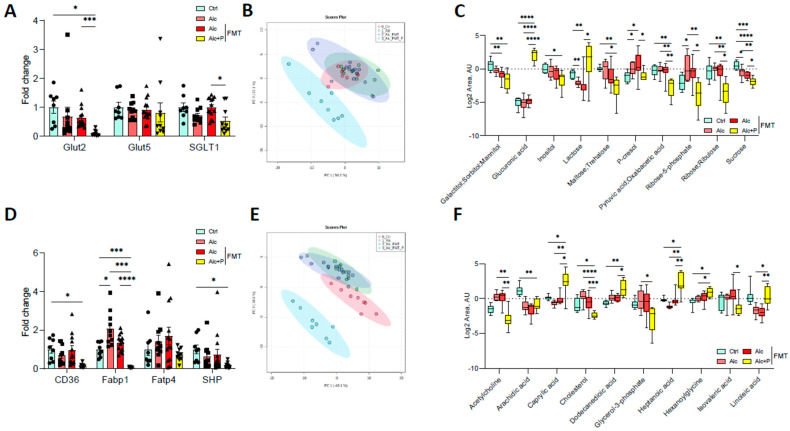
Alterations in sugar and lipid metabolism induced by pectin. Control (Ctrl) and alcohol-fed (Alc) mice received a Lieber DeCarli diet with isocaloric maltodextrin or alcohol, respectively. Pectin-treated mice (Alc + *p*) received 6.5% pectin in addition to 5% ethanol. Human fecal microbiota transfer (FMT) groups received the microbiota of a patient with alcoholic hepatitis. Quantification of mRNA levels of glucose transporters (glucose transporter 2 and 5, Glut2, Glut5; Sodium/glucose cotransporter 1, SGLT1) (**A**) and lipid transporters (cluster of differentiation 36, CD36; fatty acid binding protein 1, Fabp1; fatty acid transporter 4, Fatp4; small heterodimer partner, SHP) (**D**) by qPCR. Principal component analysis of fecal glucose-metabolites (**B**) and fecal lipid metabolites (**E**). Carbohydrates (**C**) and lipid (**F**) metabolite quantification in feces. * *p* < 0.05, ** *p* < 0.01, *** *p* < 0.001, **** *p* < 0.0001.

**Figure 7 cells-11-00968-f007:**
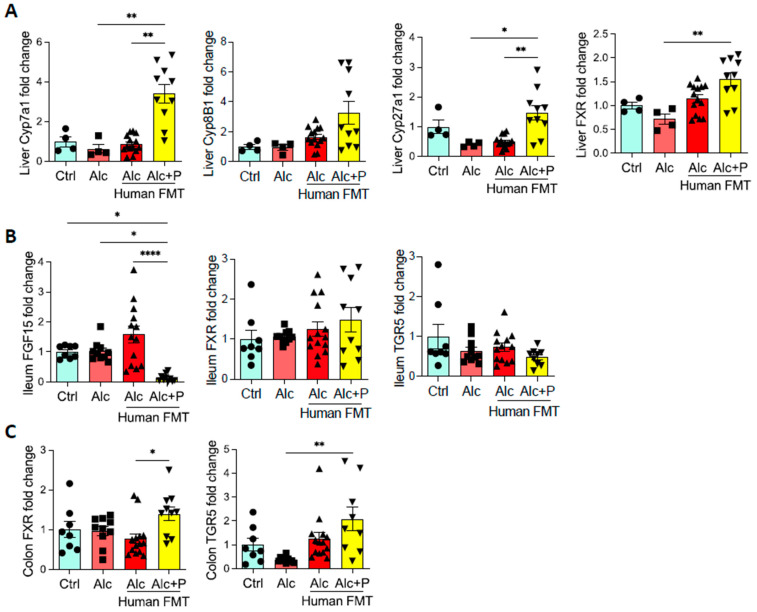
Bile acid (BA) signaling. Control (Ctrl) and alcohol-fed (Alc) mice received a Lieber DeCarli diet with isocaloric maltodextrin or alcohol, respectively. Pectin-treated mice (Alc + *p*) received 6.5% pectin in addition to 5% ethanol. Human fecal microbiota transfer (FMT) groups received the microbiota of a patient with alcoholic hepatitis. Quantification of the mRNA levels of enzymes involved in liver BA synthesis (Cholesterol 7 alpha-hydroxylase, Cyp7a1; Sterol 12-alpha-hydroxylase, Cyp8b1; sterol 27-hydroxylase, Cyp27a1; Farnesoid X receptor, FXR) (**A**), ileal BA signaling (fibroblast growth factor 15, FGF15, FXR, G protein-coupled bile acid receptor 1, TGR5) (**B**), and colon BA signaling (FXR, TGR5) (**C**) by qPCR. * *p* < 0.05, ** *p* < 0.01, **** *p* < 0.0001.

**Figure 8 cells-11-00968-f008:**
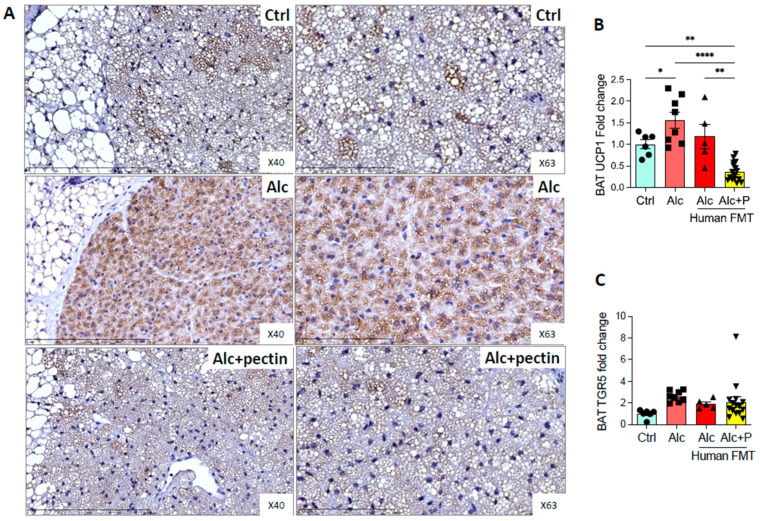
Brown adipose tissue activation in alcohol-fed mice is reversed by pectin. Control (Ctrl) and alcohol-fed (Alc) mice received a Lieber DeCarli diet with isocaloric maltodextrin or alcohol, respectively. Pectin-treated mice (Alc + *p*) received 6.5% pectin in addition to 5% ethanol. Human fecal microbiota transfer (FMT) groups received the microbiota of a patient with alcoholic hepatitis. Representative histological images of brown adipose tissue (scale bar = 100 µm and 200 µm) by uncoupling protein 1, UCP1 immunohistochemistry (**A**). Quantification of mRNA levels of UCP1 (**B**) and TGR5 (**C**) of brown adipose tissue by qPCR. * *p* < 0.05, ** *p* < 0.01, **** *p* < 0.0001.

**Figure 9 cells-11-00968-f009:**
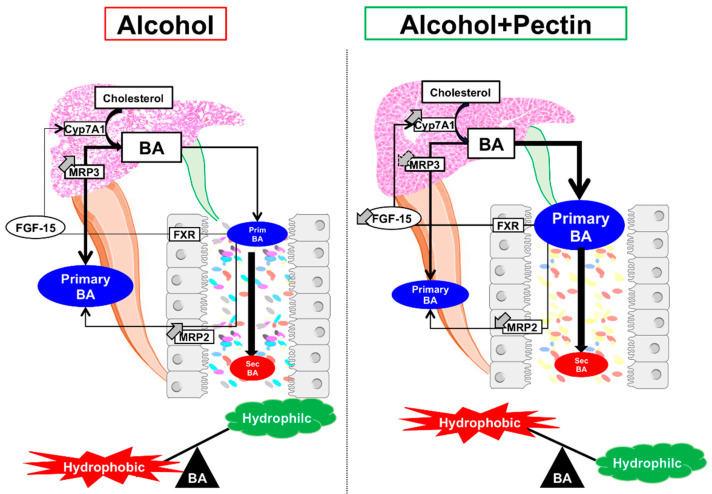
Graphical summary of the role of pectin in the bile-acid (BA) enterohepatic cycle in alcoholic liver disease. Pectin alleviates alcohol liver disease by increasing BA excretion and modifying the intestinal microbiota and the BA pool.

## Data Availability

Data is available upon request to the corresponding authors.

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
