# Peer review of "Modulation of the Bile Acid Enterohepatic Cycle by Intestinal Microbiota Alleviates Alcohol Liver Disease"

_cells, 2022, doi:10.3390/cells11060968_

Round 1
Reviewer 1 Report
Overview:
In this study, the authors aimed to elucidate how oral supplementation of pectin affects BA metabolism in alcohol-challenged mice receiving feces from patients with alcoholic hepatitis. The authors demonstrated that pectin reduced alcohol-induced liver injury. This beneficial effect correlated with lower BA levels in the plasma and liver but higher levels in the caecum, suggesting that pectin stimulated BA excretion. Pectin alleviates alcohol-induced liver injury by modifying the BA cycle through effects on the intestinal microbiota and enhanced BA excretion. Overall, the data provided here by the authors could have interesting. Specific points of view need to be considered are listed below:
- In Figure 1C, the authors showed representative histological images of Oil-Red-O staining of the liver. However, it’s necessary to quantify the positive staining and include in the figure.
- As the authors has showed in previous publications, it would be interesting to include the treatment group of Alc mice treated with FMT from a healthy donor and compare with the Alc mice treated with FMT from the alcoholic patient with severe alcoholic hepatitis.
- Besides liver mRNA levels of pro-inflammatory cytokines and chemokines, the authors should evaluate the expression of inflammatory markers by immunostaining.
- To make the study more rigorous, it’s better to measure the protein expression of FXR/BA signaling in liver, ileal and colon.
- The UCP1 immunostaining data for the Alc FMT group is missing in Figure 8A.
Author Response
We are grateful to the reviewer for his precious input and suggestions. Please find below our answers to the points they raised.
- In Figure 1C, the authors showed representative histological images of Oil-Red-O staining of the liver. However, it’s necessary to quantify the positive staining and include in the figure.
We agree that usually the Oil-Red-O staining is presented with its quantification. However we choose to measure the quantity of triglycerides in the liver directly (Fig 1B), which is a more accurate (quantitative vs. semi quantitative for the quantification of the positive Oil-Red-o staining) and well-accepted method in the literature to estimate steatosis.
2. As the authors has showed in previous publications, it would be interesting to include the treatment group of Alc mice treated with FMT from a healthy donor and compare with the Alc mice treated with FMT from the alcoholic patient with severe alcoholic hepatitis.
In our previous publications we used Alc mice treated with FMT from patients without severe ALD to prove that the intestinal microbiota from patients with severe alcoholic hepatitis worsen liver injury in the mouse model. Moreover, we showed that modifying this deleterious IM in sAH FMT mice using pectin we can alleviate ALD. However, in the present study we wanted to gain further insights in the mechanism related to how pectin works in the context of the deleterious microbiota from patients with sAH.
Although interesting, including a group of mice that will receive IM from healthy donors won’t increase the understanding of the physiology of sAH and it is not reasonable from an ethics point of view.
3. Besides liver mRNA levels of pro-inflammatory cytokines and chemokines, the authors should evaluate the expression of inflammatory markers by immunostaining.
We agree that it would be interesting to couple the liver mRNA expression of pro-inflammatory cytokines with immunostaining of inflammatory markers. However in the Liber De Carli model that we use, liver mRNA levels of pro-inflammatory cytokines and chemokines are a standard and well-accepted way of estimating inflammation and it is largely used in mice studies in the academic community (eg. studied from the groups of Bernd Schnabl, Bin Gao, Gyongyi Szabo, etc...).
We could perform F4/80 immunostaining to quantity macrophages in the liver but the delay of the editor of 10 days is too short to do that. If the reviewer considers that this point is essential we would kindly ask the editor to give us more time in order to be able to perform this specific labelling.
4. To make the study more rigorous, it’s better to measure the protein expression of FXR/BA signalling in liver, ileal and colon.
We agree with the reviewer that protein expression would be a valuable tool for our study. Unfortunately we do not have any more ileal and colon samples to perform protein expression including FGF15 in these compartments.
For the liver we would usually quantify Cyp7a1 expression to evaluate the FXR/BA signalling pathway. However, FXR is a bile acid-activated transcription factor who orchestrates BA synthesis and the enterohepatic BA cycle. The transcription of CYP7a1 is directly regulated through the transcription activation of FXR, therefore, mRNA expression is a good surrogate of the protein level.
However, if you consider that performing at least CYP7a1 expression in the liver is essential we would need more time to perform these experiments.
5. The UCP1 immunostaining data for the Alc FMT group is missing in Figure 8A.
We thank the reviewer for this point, as there was a mistake in our figure. In the immunostainings that we present (Fig 8A) we only have the Ctr, Alc and Alc+Pectin groups. The UCP1 immunostaining and mRNA expression are from two different experiments. In the first experiment we recovered BAT for mRNA expression (Fig 8B). As the results were promising we harvest the BAT for immunostaining in a second experiment where we only included the Ctrl, Alc and Alc+pectin groups to support the results of mRNA expression. We corrected the figure accordingly.
Reviewer 2 Report
The manuscript by Ciocan et al on how the modulation of the bile acid enterohepatic cycle by IM alleviates alcohol liver disease is a well-performed study using state of the art models such as humanized microbiota animal models, these models can offer a better understanding of the associations between microbiota and ALD. In my concern it is not very clear why the authors have combined the mouse Alc model with the human FMT and why seldom they tend on to perform comparisons between pectin treatment on the Alc + human FMT model with the mouse Alc model without the human FMT. Maybe pectin treatment should have been carried out also in the mouse Alc model and assess if pectin treatment also induced changes in the BA composition as it does with the Alc + human FMT model.
Another interesting point is the fact that even though human FMT transplant in the Alc mouse model is associated with increased ALD, the total BA composition and the BA composition is not significantly altered in liver between these two models. Can you comment on this?
Other major concerns include:
In Fig. 1, the authors state that FMT from patients with sAH to mice worsened alcohol-induced liver injury in the LDC model of ALD. However, no significant differences are shown in liver steatosis and only on pro-inflammatory gene (CCl2) is augmented between Alc and Alc + human FMT.
In Figure 2, the authors state that mass spectrometry analysis showed changes in the overall composition of the BA pool in plasma, the liver and caecum. However, the PCA analysis performed is not so clear and some statistic data should be shown to clarify the separation between the different groups. Moreover, the bad resolution of the figure does not allow to see very well the points of each group in the PCA graph. On line 222, the authors focus on the fact that pectin treatment resulted in lower plasma TBA levels than in Alc mice that did not receive pectin, but the comparison should be with the Alc + human FMT where no difference occurs. In fact, no significant difference between pectin treatment on Alc + human FMT is detected in TBA in plasma, liver, and caecum, according to Figs. 2A-C. The differences are observed when comparing the proportion of conjugated and unconjugated BAs.
In Figure 3, regarding the impact of pectin on the composition of BA, once again the authors should center in the differences between the group of Alc + human FMT and Alc + human FMT + pectin and under these conditions the major findings are observed in the caecum, a reduction of conjugated BA in the plasma whereas in the liver only differences in the TDCA are detected. Therefore, the sentence in 247-249 is not very accurate and should be rewritten.
Other minor points include:
Statistical analysis for all the PCA data is missing
Abbreviations in figure legends are missing.
Author Response
We are grateful to the reviewer for his precious input and suggestions. Please find below our answers to the points he raised.
- In my concern it is not very clear why the authors have combined the mouse Alc model with the human FMT and why seldom they tend on to perform comparisons between pectin treatment on the Alc + human FMT model with the mouse Alc model without the human FMT.
We understand that it may be confusing using the two groups: Alc and Alc + human FMT model. In our previous studies we have showed that, the microbiota from patients with severe alcoholic hepatitis worsen liver lesions in the Lieber De Carli model (Llopis M. et al Gut 2016; Wrzosek L. et al, Gut 2021). We present always the Alc group to show that our model is reproducible and that even if we use intestinal microbiota from different patients with sAH we have the same effect in the model: worsen alcohol induced liver lesions.
- Maybe pectin treatment should have been carried out also in the mouse Alc model and assess if pectin treatment also induced changes in the BA composition as it does with the Alc + human FMT model.
We understand the point that the reviewer is raising. However, our work focuses on a human disease, severe alcoholic hepatitis and the role of intestinal microbiota. There is no relevance to show that changing murine microbiota can alleviate alcohol liver disease, as severe alcoholic hepatitis is not a murine disease.
- Another interesting point is the fact that even though human FMT transplant in the Alc mouse model is associated with increased ALD, the total BA composition and the BA composition is not significantly altered in liver between these two models. Can you comment on this?
We agree that the total BA level is not significantly modified in liver between the two models. However, we observed an increase in the total BA in FMT model. The lack of a statistical significant difference may be do to a lack of power. Moreover, as we compared four groups we performed adjustments for multiple comparisons, which could also explain the lack of power. We did not determined the number of mice needed for this study based on this criterion and therefore the study was not designed to test any different in bile acids quantity.
However when we looked at the primary conjugated BA in the liver, there was an increase in these Bas in the FMT model as compared to Alc model. Nevertheless, when analysing the different species of Bas there was no difference between Alc and Alc FMT models.
These data suggests that in the presence of Alc, FMT increases the production of BAs (increase in primary conjugates BAs), however as the BAs are rapidly removed from the liver we may not be able to see small changes in this compartment.
- In Fig. 1, the authors state that FMT from patients with sAH to mice worsened alcohol-induced liver injury in the LDC model of ALD. However, no significant differences are shown in liver steatosis and only on pro-inflammatory gene (CCl2) is augmented between Alc and Alc + human FMT.
We agree that there are markers of liver injury that do not vary between Alc and Alc FMT model. However, the ALT (the most common marker of liver damage) and some direct (Ccl2) and indirect markers of inflammation (Nos2) are significantly increased between Alc and Alc FMT model. Other inflammatory markers such as TNFα, Il1β and Ccl3 show a tendency to increase in Alc FMT vs Alc mice. Therefore, we considered that overall the Alc FMT model was associated with more severe liver damage in our model.
- In Figure 2, the authors state that mass spectrometry analysis showed changes in the overall composition of the BA pool in plasma, the liver and caecum. However, the PCA analysis performed is not so clear and some statistic data should be shown to clarify the separation between the different groups.
The PCA is only used to show clusters of samples based on their bile acids similarity. There are no common statistics used to compare groups using PCA. When we used the syntax “changes in the overall composition of the BA pool in plasma, the liver and caecum” we were referring to the total BA quantity which also reflects the composition of the BA pool.
- Moreover, the bad resolution of the figure does not allow to see very well the points of each group in the PCA graph.
We tried to improve the resolution of the image.
- On line 222, the authors focus on the fact that pectin treatment resulted in lower plasma TBA levels than in Alc mice that did not receive pectin, but the comparison should be with the Alc + human FMT where no difference occurs. In fact, no significant difference between pectin treatment on Alc + human FMT is detected in TBA in plasma, liver, and caecum, according to Figs. 2A-C. The differences are observed when comparing the proportion of conjugated and unconjugated BAs.
We agree with the reviewer’s comment and corrected the text as follows:
“Conversely, pectin treatment resulted in lower plasma TBA levels than in Alc mice that did not receive pectin (Fig. 2A) and showed a tendency towards lower plasma (Fig. 2B) and liver TBA (Fig. 2C), and higher caecal TBA (Fig. 2C) levels as compared to alcohol-fed FMT mice, without reaching statistical significance.” (Lines 181-184).
- In Figure 3, regarding the impact of pectin on the composition of BA, once again the authors should center in the differences between the group of Alc + human FMT and Alc + human FMT + pectin and under these conditions the major findings are observed in the caecum, a reduction of conjugated BA in the plasma whereas in the liver only differences in the TDCA are detected. Therefore, the sentence in 247-249 is not very accurate and should be rewritten.
We agree with the reviewer that the mains interest in terms of changes in Bas is between the group of Alc + human FMT and Alc + human FMT + pectin groups and that it may be difficult to follow all the changes between all the groups.
As suggested by the reviewer we changes our paragraph as follows in order to simplify and centre our results and message on what is important:
“Conversely, pectin treatment resulted in lower primary conjugated BA levels in the plasma (TCA and TMCA) than in conventional and FMT-treated Alc mice, whereas they remained as high as in the controls (Fig. 2D, Fig. 3A). In the caecum, the alcohol diet re-sulted in lower primary (CA, CDCA, bMCA) and secondary unconjugated BA (DCA, UDCA, wMCA) levels than in control mice, whereas pectin treatment resulted in higher primary unconjugated BA levels (CA and CDCA) than in conventional and FMT-treated Alc mice and a further decrease in secondary unconjugated BA levels (DCA, LCA, UDCA, wMCA) (Fig. 2F, Fig. 3C). Pectin treatment also resulted in higher primary conjugated BA (TCA, TMCA) and TUDCA levels in the caecum than in all other groups, notably FMT-treated Alc mice (TCA, TMCA, TUDCA) (Fig. 2F, Fig.3C). Overall, pectin decreased plasma and liver BA levels, increased caecal BA levels, mainly those of primary BAs, CA, and MCA, and changed the overall composition of the BA pool towards more hydrophilic forms (Fig 3D, E).” (Lines 201-211).
- Other minor points include: Statistical analysis for all the PCA data is missing
Principal component analysis (PCA) is an exploratory statistical method for graphical description of the information present in large datasets. The aim is to synthesize the huge quantity of information (in our case the different species of bile acids) into an easy and understandable form. In the figure we present the variance explained by the first two components of the PCA (on the X and Y axis), which is the parameter that we take into account to estimate the relevance of the representation. However we understand that the resolution could be a problem and therefore we included these metrics in the figure legends.
- Abbreviations in figure legends are missing.
Thank you for pointing this omission. We added the abbreviations in the figure legends as suggested.
Reviewer 3 Report
GENERAL COMMENT
A recent study found that, in humans, there are distinct changes in BA-transforming microbiota and corresponding BAs in alcohol-associated hepatitis that are related to disease severity (Hepatol Commun. 2022 Jan 4. doi: 10.1002/hep4.1885. Epub ahead of print. PMID: 34984859). This study attests to the importance of addressing the triangular association of Bas, gut microbiota and alcohol-associated liver disease. With this backset, the submission entitled "Modulation of the bile acid enterohepatic cycle by intestinal microbiota alleviates alcohol liver disease" by Dr Ciocan and Colleagues is a valuable and timely contribution. I enjoyed reading this study and I have some suggestions aimed at improving this submission.
SPECIFIC COMMENT
1. Throughout the manuscript the same condition is variably defined, i.e. "alcohol liver disease” (Title); “ alcohol-induced liver injury” (Abstract); and “ Alcohol-associated liver disease (ALD) (Background)”. I suggest to be consistent.
2. Keywords - "Keywords: keyword 1; keyword 2; keyword 3 (List three to ten pertinent keywords specific to the article yet reasonably common within the subject discipline.) " This is not acceptable
3. Introduction is not particularly straightforward to follow and must be reworked. I have some suggestions aimed at improving this. For example: rework this section into three definite sections:
a) ALD – here notions regarding definition, classification, epidemiological and clinical burden should be summarized.
b) Gut microbiota – Definition and general physiology with specific reference to BAs
c) Interaction of Gut Microbiota with alcohol metabolism; role of Bas; rationale and aims.
Background is not a summary/abstract, therefore the following statement must be deleted “We found that pectin, acting as a sequestrant of BAs, switches their overall composition towards more hydrophilic and consequently less-toxic species, thus decreasing liver injury.“. Conversely, the rationale of the study must be argued more extensively. To this end, sections a) and b) should include all information to explain what – based on previous studies - these authors expected to find and why.
I also suggest to be more accurate and to support with appropriate bibliographic references all relevant statements, for example: “ALD includes various histopathological characteristics, ranging from simple steatosis to steatohepatitis, fibrosis, and cirrhosis, but only 10 to 20% of patients evolve to advanced fibrosis and its complications. In addition, certain ALD patients develop alcoholic hepatitis, with episodes of acute liver inflammation associated with high mortality and few therapeutic options. “
The following statement should not anticipate the section on Gut Microbiota but may be included in the rationale of the study.“Interventions on the IM by fecal microbial transfer or prebiotic or probiotic treatments alleviate alcohol-induced liver injury in animal models. Microbiota-editing strategies could therefore constitute a promising therapeutic approach for the treatment of ALD [3].“
Lines 52 to 90 must be summarized to a substantial extent.
4. Method – Recently, the role of sex as a major modifier of health and disease has gained attention. In this setting I suggest to explain (Background) what the features of alcoholic liver disease in humans are regarding sex differences (Lancet. 2020 Aug 22;396(10250):565-582). Such notions should be used to explain why female animals were utilized.
5. Discussion
a) – Lines 363-376 may be used to explain the rationale of the study (Background) and Discussion should start by summarizing novel findings.
b) It is widely acknowledged that the distinction of alcoholic from nonalcoholic liver disease is somewhat artificial (World J Gastroenterol. 2012 Jul 21;18(27):3492-501 Curr Pharm Des. 2020;26(10):1093-1109) and indeed the novel MAFLD concept tends to highlight this. With this backset, I suggest to discuss that fibers (including pectin) do play a role in the setting of energy homeostasis and personalized medicine approaches to metabolic disorders (Nutrients. 2021 Sep 29;13(10):3470. Metab Target Organ Damage 2021;1:3. http://dx.doi.org/10.20517/mtod.2021.03. Metab Target Organ Damage 2021;1:9. http://dx.doi.org/10.20517/mtod.2021.11 ).
c) This study found no effect of pectin on TGR5 mRNA levels in the BAT of alcohol-fed mice. In this regard the following study should be discussed: Metab Target Organ Damage 2021;1:8. http://dx.doi.org/10.20517/mtod.2021.04
6. References
All the above references, including the study by Muthiah (Hepatol Commun. 2022 Jan 4. doi: 10.1002/hep4.1885. Epub ahead of print. PMID: 34984859) should be addressed.
Author Response
We are grateful to the reviewer for his precious input and suggestions. Please find below our answers to the points he raised.
- Throughout the manuscript the same condition is variably defined, i.e. "alcohol liver disease” (Title); “ alcohol-induced liver injury” (Abstract); and “ Alcohol-associated liver disease (ALD) (Background)”. I suggest to be consistent.
We agree that the use of different syntax for the same condition may be misleading. Therefore we harmonized the manuscript using “alcohol liver disease”.
- Keywords - "Keywords: keyword 1; keyword 2; keyword 3 (List three to ten pertinent keywords specific to the article yet reasonably common within the subject discipline.) " This is not acceptable
We apologize for this omission that we corrected in the revised version of the manuscript.
- Introduction is not particularly straightforward to follow and must be reworked. I have some suggestions aimed at improving this. For example: rework this section into three definite sections:
- a) ALD – here notions regarding definition, classification, epidemiological and clinical burden should be summarized.
- b) Gut microbiota – Definition and general physiology with specific reference to BAs
- c) Interaction of Gut Microbiota with alcohol metabolism; role of Bas; rationale and aims.
We thank the reviewer for his precious remarks and suggestions that we used to improve our manuscript.
- Background is not a summary/abstract, therefore the following statement must be deleted “We found that pectin, acting as a sequestrant of BAs, switches their overall composition towards more hydrophilic and consequently less-toxic species, thus decreasing liver injury.“.
We deleted this part.
- Conversely, the rationale of the study must be argued more extensively. To this end, sections a) and b) should include all information to explain what – based on previous studies - these authors expected to find and why.
We thank the reviewer for this comment but we consider that we argued quite extensively, why we wanted to study the role of pectin on the BA cycle. We used your remark from below and used the Lines 363-376 (from our first submission) to better explain our rationale.
- I also suggest to be more accurate and to support with appropriate bibliographic references all relevant statements, for example: “ALD includes various histopathological characteristics, ranging from simple steatosis to steatohepatitis, fibrosis, and cirrhosis, but only 10 to 20% of patients evolve to advanced fibrosis and its complications. In addition, certain ALD patients develop alcoholic hepatitis, with episodes of acute liver inflammation associated with high mortality and few therapeutic options. “
We added the references lacking.
- The following statement should not anticipate the section on Gut Microbiota but may be included in the rationale of the study.“Interventions on the IM by fecal microbial transfer or prebiotic or probiotic treatments alleviate alcohol-induced liver injury in animal models. Microbiota-editing strategies could therefore constitute a promising therapeutic approach for the treatment of ALD [3].“
We agree that this statement should be included in the rationale of the study and we moved it accordingly.
- Lines 52 to 90 must be summarized to a substantial extent.
We thank the reviewer for this comment but we consider that it is essential for the readership to have a reminder of bile acid physiology and its link to intestinal microbiota. This is crucial for the understanding of the relevance of our work in the context of alcohol liver disease.
- Method – Recently, the role of sex as a major modifier of health and disease has gained attention. In this setting I suggest to explain (Background) what the features of alcoholic liver disease in humans are regarding sex differences (Lancet. 2020 Aug 22;396(10250):565-582). Such notions should be used to explain why female animals were utilized.
Indeed sex is a well-recognized independent risk factor of ALD for decades. In mice, as in humans, females are more susceptible to develop liver injury than males for a similar amount of alcohol intake. As suggested, we added the reference kindly provided (Lines 41-43) and we detailed the choice of female mice (Lines 104-105).
- Discussion
- a) – Lines 363-376 may be used to explain the rationale of the study (Background) and Discussion should start by summarizing novel findings.
We agree with reviewer and we modified the text accordingly.
b) It is widely acknowledged that the distinction of alcoholic from nonalcoholic liver disease is somewhat artificial (World J Gastroenterol. 2012 Jul 21;18(27):3492-501 Curr Pharm Des. 2020;26(10):1093-1109) and indeed the novel MAFLD concept tends to highlight this. With this backset, I suggest to discuss that fibers (including pectin) do play a role in the setting of energy homeostasis and personalized medicine approaches to metabolic disorders ( 2021 Sep 29;13(10):3470. Metab Target Organ Damage 2021;1:3. http://dx.doi.org/10.20517/mtod.2021.03. Metab Target Organ Damage 2021;1:9. http://dx.doi.org/10.20517/mtod.2021.11 ).
The new concept of MAFLD to designate ALD is still debated in the liver specialist community while it is more easily accepted as an upgraded term for NAFLD (Devi J,. World J Hepatol. 2022 Jan 27;14(1):158-167. doi: 10.4254/wjh.v14.i1.158., ). Although both ALD and NAFLD share common histological findings there are several major differences in term of pathology and prognostic. The same is true for the role and the mechanisms by which the intestinal microbiota modulates these two diseases with some common points but also some important differences (reviewed in Cell Host Microbe. 2020 Aug 12;28(2):233-244.doi: 10.1016/j.chom.2020.07.007.).
Our study focused only on ALD and we did not perform any model of NAFLD, therefore discussing the role of fibres and personalized medicine approaches in metabolic disorders would go far beyond the scope and the results of our study and would be pure hypothesis. It would be indeed interesting to address these points in a review or a perspective paper.
- c) This study found no effect of pectin on TGR5 mRNA levels in the BAT of alcohol-fed mice. In this regard the following study should be discussed: Metab Target Organ Damage 2021;1:8. http://dx.doi.org/10.20517/mtod.2021.04
We thank the reviewer for this suggestion however in the above-mentioned study the authors’ studied TGR5 expression in visceral adipose tissue, which is a white adipose tissue, and in the setting of obesity. In our case, we are in the context of ALD and in the brown adipose tissue. Therefore, we do not consider that discussion this interesting paper will help the understanding of our results.
- References
All the above references, including the study by Muthiah (Hepatol Commun. 2022 Jan 4. doi: 10.1002/hep4.1885. Epub ahead of print. PMID: 34984859) should be addressed.
We kindly thank the reviewer for his suggestions that we tried our best to integrate in our manuscript.
Round 2
Reviewer 1 Report
My comments were not addressed.
Reviewer 2 Report
The authors have addressed my comments and as far as I am concerned the manuscript is now suitable for publication.